# Keratinocyte Response to Infection with *Sporothrix schenckii*

**DOI:** 10.3390/jof8050437

**Published:** 2022-04-23

**Authors:** Araceli Paredes-Rojas, Alejandro Palma-Ramos, Laura Estela Castrillón-Rivera, Felipe Mendoza-Pérez, María del Carmen Navarro-González, Roberto Arenas-Guzmán, Jorge Ismael Castañeda-Sánchez, Julieta Luna-Herrera

**Affiliations:** 1Departamento de Inmunología, Escuela Nacional de Ciencias Biológicas, Instituto Politécnico Nacional, Mexico City 11340, Mexico; aracelibsb14@gmail.com; 2Departamento de Sistemas Biológicos, Universidad Autónoma Metropolitana Unidad Xochimilco, Mexico City 04960, Mexico; alpalma@correo.xoc.uam.mx (A.P.-R.); lcrivera@correo.xoc.uam.mx (L.E.C.-R.); fmendoza@correo.xoc.uam.mx (F.M.-P.); 3Laboratorio de Investigación en Enfermedades Reumáticas, Instituto Nacional de Enfermedades Respiratorias “Ismael Cosío Villegas”, Mexico City 14080, Mexico; carmen.navarro@iner.gob.mx; 4Sección de Micología, Hospital General “Dr. Manuel Gea González”, Mexico City 14080, Mexico; rarenas98@hotmail.com

**Keywords:** *Sporothrix schenckii*, Sporotrichosis, keratinocytes, fungal infection, innate immunity, inflammation, cytoskeleton, toll-like receptors, cytokines

## Abstract

Sporotrichosis is a subacute, or chronic mycosis caused by traumatic inoculation of material contaminated with the fungus *Sporothrix schenckii* which is part of the *Sporothrix* spp. complex. The infection is limited to the skin, although its progression to more severe systemic or disseminated forms remains possible. Skin is the tissue that comes into contact with *Sporothrix* first, and the role of various cell lines has been described with regard to infection control. However, there is little information on the response of keratinocytes. In this study, we used the human keratinocyte cell line (HaCaT) and evaluated different aspects of infection from modifications in the cytoskeleton to the expression of molecules of the innate response during infection with conidia and yeast cells of *Sporothrix schenckii*. We found that during infection with both phases of the fungus, alterations of the actin cytoskeleton, formation of membrane protuberances, and loss of stress fibers were induced. We also observed an overexpression of the surface receptors MR, TLR6, CR3 and TLR2. Cytokine analysis showed that both phases of the fungus induced the production of elevated levels of the chemokines MCP-1 and IL-8, and proinflammatory cytokines IFN-α, IFN-γ and IL-6. In contrast, TNF-α production was significant only with conidial infection. In late post-infection, cytokine production was observed with immunoregulatory activity, IL-10, and growth factors, G-CSF and GM-CSF. In conclusion, infection of keratinocytes with conidia and yeast cells of *Sporothrix schenckii* induces an inflammatory response and rearrangements of the cytoskeleton.

## 1. Introduction

The Sporotrichosis is a mycosis that involves epidermis, dermis and subcutanous tissue and is caused by thermally dimorphic fungal species of the complex *Sporothrix* spp. One of its etiological agents is the fungus *Sporothrix schenckii* [1,2]. This fungus is found in mycelial form in the environment (saprophytic or infectious phase) at a temperature of 20–30 °C and enters the skin by traumatic inoculation of propagules that infect the skin and subcutaneous tissue, developing into yeast form (parasitic phase) at 37 °C [3,4]. Conversion of the mycelial form to yeast form is necessary for the infection that causes the most common clinical forms, lymphocutaneous and fixed cutaneous, to be established [5]. 20–30% of the cases have a fixed presentation, which is characterized by a nodular lesion called sporotrichoid chancre. The lymphangitic form, on the other hand, presents with ulcerated nodules that follow a linear path along the lymphatic vessels in the extremities. Without treatment, affected people may develop a disseminated (extracutaneous) disease. Extracutaneous forms are rare and affect bones and joints [5].

Sporotrichosis is a globally distributed mycosis. Most cases occur in tropical and subtropical areas; in Latin America, countries such as Brazil, Colombia, El Salvador, Mexico, Uruguay, and Venezuela have an estimated prevalence rate of 0.1 to 0.5%, while in Mexico, the largest number of cases reported are in Jalisco, Mexico City, Puebla, Guerrero and Guanajuato [6,7,8]. In recent years, it has been considered an emerging mycosis due to multiple outbreaks, its taxonomic evolution, and its wide distribution [8].

The innate immune response is actively involved in controlling this mycosis. It has been demonstrated in murine models with chronic granulomatous disease that mechanisms based on reactive oxygen species (ROS) are essential for the elimination of yeast cells phagocytized by neutrophils and macrophages [9]. On the other hand, *S. schenckii* yeast cells activate the alternative complement pathway that allows their phagocytosis [10]. Dectin-1 and the mannose receptor (MR) are among those receptors described as being involved in the recognition of this pathogen [11,12]. Opsonized conidia of *S. schenckii* have also been shown to be efficiently phagocytized by THP-1 macrophages [11]. On the other hand, human peripheral blood mononuclear cells (PBMCs) co-cultured for 24 h with conidia and yeast cells of *S. schenckii* stimulated similar levels of the proinflammatory cytokines TNF-α, IL-6, andIL-1β. In addition, blocking the MR had an effect on TNF-α, IL-6, and IL-10 levels [12].

The skin is the main tissue that gets in contact with potentially pathogenic microorganisms such as *S. schenckii* and keratinocytes are the most abundant cells in the epidermis. Keratinocytes have essential roles in restoring the epidermal barrier and have recently received attention for their role in initiating immune responses [13,14]. Keratinocytes express pathogen recognition receptors (PRRs) such as toll-like receptors (TLRs) from TLR-1 to TLR-6 and TLR-9, retinoic acid-inducible receptors type 1 (RIG-1), C-type lectin receptors such as Dectin-1, and nucleotide-binding oligomerization domain-containing protein 1 (NOD-1) [15,16,17,18,19,20]. The activation of TLRs induces the expression of pro-inflammatory cytokines such as IL-8 and antimicrobial peptides (AMPs). Antimicrobial peptides are an important component in the skin’s immune function [15]. Keratinocytes have been described as playing a role in the control of infections by various pathogens [21]. However, little is known about the interaction of keratinocytes with *S. schenckii*. This study aimed to evaluate the response of human keratinocytes infected with conidia and yeast cells of *S. schenckii*, by analyzing changes in the distribution of the actin cytoskeleton, expression of surface receptors, production of cytokines and chemokines, and production of growth factors, at various times of infection from keratinocytes.

## 2. Materials and Methods

### 2.1. HaCaT Keratinocyte Culture

The HaCaT human keratinocyte cell line (Thermo Fisher Scientific, Waltham, MA, USA), was maintained in F-12 medium (Gibco by Life Technologies, Grand Island, NY, USA) supplemented with 10% fetal bovine serum (SFB) (Gibco by Life Technologies, Grand Island, NY, USA), and a 1% antibiotic mixture of streptomycin and penicillin (Gibco by Life Technologies). The cells were kept at 37 °C in a 5% CO_2_ atmosphere.

### 2.2. Sporothrix schenckii Culture

A clinical isolate of *Sporothrix schenckii* was used, recovered from a patient with disseminated sporotrichosis treated in Hospital General “Dr. Manuel Gea Gonzalez”, SSA, Mexico. The culture was characterized using microbiological and molecular methods [5]. The conidial phase was obtained by sowing the fungus in a Sabouraud Dextrose agar medium (Eur Pharm, Madrid, Spain) and incubating it at 25 °C for 30 days. Fungal yeasts were obtained in BHI agar medium (Applichem Panreac, Darmstadt, Germany) at 37 °C for 5 days. Suspensions were prepared with both phases of the fungus by taking a colony of each phase and adjusting them in a sterile saline solution to reach 300 × 10^6^ CFU/mL via the nephelometric method.

### 2.3. Cell Infection

Monolayers of keratinocytes were prepared in 12-well plates at a concentration of 400 × 10^3^ cells per well. They were infected for 2 h with a suspension of conidia or yeasts with a 1:1 multiplicity of infection (MOI). After 2 h of infection, the monolayers were washed with 1X PBS, and samples were taken at 2, 4, 6, 8, 10, and 12 h following infection. Uninfected cells were used as negative controls for all cases. To prevent extracellular growth of *Sporothrix*, the infected monolayers were treated with 1 μg/mL amphotericin B (Thermo Fisher Scientific).

### 2.4. Cytotoxicity Analysis

Cell infection kinetics were analyzed, and cell culture supernatants were recovered at each post-infection time. The cytotoxicity percentage of the keratinocytes was determined following the protocol of the CytoTox 96^®^ Non-Radioactive Cytotoxicity Assay (Promega, Madison, WI, USA) at each post-infection time, and uninfected cells were used as a negative control.

### 2.5. Analysis of Changes in the Cytoskeleton of Infected Keratinocytes

Keratinocyte monolayers were prepared as described above, but on sterile coverslips placed on 12-well plates, and infected with the conidia or yeast suspensions in a manner similar to that described above. Changes in actin distribution were analyzed at 2, 6, and 10 h post-infection. At each time, cells were fixed with 4% paraformaldehyde for 30 min, washed with 1X PBS, and stained with 80 ng Rhodamine (TRITC) Phalloidin (Sigma-Aldrich, Steinheim, Germany) for 20 min. Excess Phalloidin was removed through five washes with 1X PBS. Subsequently, the samples were mounted on slides using Vectashield-DAPI (Vector Laboratories, Inc., Burlingame, CA, USA).

### 2.6. Opsonization of Conidia and Yeast Cells of S. schenckii

Suspensions of 30 × 10^6^ conidia or yeasts were opsonized with 1 mL of fresh human serum for 30 min at 37 °C. Excess serum was removed by washing with 1X PBS, and the opsonized conidia and yeasts suspensions were adjusted to 300 x10^6^ CFU/mL in F-12 medium.

### 2.7. Analysis of Surface Receptor Expression in Infected Keratinocytes

To determine the expression of Toll-like receptors (TLR2, TLR4 and TLR6), Mannose receptor (MR) and Complement receptor 3 (CR3), the keratinocyte monolayers, were infected with conidia and yeasts as described above. As a positive control, the cells were stimulated for 24 h with PMA (phorbol myristate acetate; 1 μg/10^6^ cells), and uninfected cells were used as a negative control. The preparations were fixed with 4% paraformaldehyde (Sigma-Aldrich) for 30 min and washed three times with 1X PBS. All samples were incubated for 30 min at room temperature with a 3% BSA solution (Sigma Aldrich). Then, 50 μL of each monoclonal antibody diluted at a 1:200 was added and allowed to react overnight at 4 °C. For each receptor tested, the following antibodies were used: mouse anti-hTLR2 IgG (Santa Cruz Biotechnology, Inc., Dallas, TX, USA), mouse anti-hTLR4 IgG (eBioscience, Santa Clara, CA, USA), goat anti-hTLR6 IgG (Santa Cruz Biotechnology, Inc.), mouse IgG-anti-hCR3 (ABCAM, Cambridge, UK). Cells were then washed with 1X PBS and incubated for 90 min at 37 °C with the mouse anti-IgG secondary antibody developed in goat TRITC-labeled (Sigma Aldrich), goat anti-IgG developed in donkey FITC labeled (Santa Cruz Biotechnology, Inc.) respectively. For determining the mannose receptor, anti-mannose IgG- FITC antibody (Santa Cruz Biotechnology, Inc.) was used. Finally, the preparations were washed with 1X PBS, and mounted on slides using Vectashield DAPI (Vector Laboratories, Inc.).

Fluorescence signals were observed in a confocal scanning system (LSM5 Pascal, Zeiss, Jena, Germany). The fluorescence intensity analysis was performed with the LSM5 program (version 4.0.0.241, Confocal Zeiss, Ostfildern, Germany). For this purpose, at least 50 cells were counted per field, and the data are shown as the mean of the fluorescence intensity (MFI).

### 2.8. Cytokine Analysis in Infected Keratinocytes

Cell supernatants from conidia and *Sporothrix schenckii* yeast infection kinetics were used for the quantification of the cytokines: IFN-α, IFN-γ, IL-1β, IL-1RA, IL-2, IL-2R, IL-4, IL-5, IL-6, IL-7, IL-8, IL-10, IL-12 (p40/p70), IL-13, IL-15, IL-17, and TNF-α. Chemokines: Eotaxin, IP-10, MCP-1, MIG, MIP-1α, MIP-1β and RANTES. Growth factors: GM-CSF, G-CSF, EGF, FGF-basic, HGF and VEGF. Quantification was performed using LUMINEX technology on Magpix^®^ equipment, following the instructions of the Cytokine Human Magnetic 30-Plex Panel for the Luminex^TM^ Platform (Thermo Fisher Scientific).

### 2.9. Statistical Analysis

In all the determinations, the data were represented as the mean ± standard deviation (SD). The determinations were performed in triplicate, except the cytokine analysis, which was performed in duplicate. Data was analyzed with the two-way ANOVA test, followed by a post-hoc Tukey test, with the statistical program GraphPad Prism version 8.0 (GraphPad Software, San Diego, CA, USA). A value of *p* < 0.05 was considered to be statistically significant.

## 3. Results

### 3.1. S. schenckii Yeast Cells Induce a Cytotoxic Effect

*S. schenckii* undergoes a morphological transition in response to temperature, and this adaptation is important for the establishment of infection. An efficient transition from conidia to yeast has an impact on its virulence [22]. To establish a possible difference in cytotoxic capacity between conidia and *S. schenckii* yeasts, keratinocytes were infected with both phases of the *S. schenckii* fungus at a 1:1 MOI, and cytotoxicity was determined by the LDH release assay as described above. At 2 and 4 h post-infection, no significant differences were observed in the percentage of cytotoxicity induced by conidia and *S. schenckii* yeasts, compared with the control group of uninfected cells. From 6, 10, and 12 h post-infection, an increase in the percentage of dead cells was observed when they were infected with yeasts of the fungus (Figure 1). Infection with conidia only showed a significant increase in cytotoxicity at 10 h post-infection.

### 3.2. Sporothrix schenckii Induces Changes in the Actin Cytoskeleton in Human Keratinocytes

Once it was established that yeasts induced a higher percentage of cell death compared to conidia, cytoskeletal rearrangements were analyzed in cells infected with both phases of the fungus. Actin filaments were stained with rhodamine phalloidin, and the cell nucleus with DAPI, and then analyzed via confocal microscopy. As shown in Figure 2, the keratinocytes that were not infected showed a homogeneous distribution of the actin filaments, without cellular prolongations and with a longitudinal organization. In contrast, keratinocytes infected with conidia of *S. schenckii* showed morphological changes starting 2 h post-infection. The cells presented a loss in the longitudinal distribution of the filaments and focal points of actin were observed throughout the cytoplasm. These changes were maintained until 10 h post-infection.

Moreover, the yeast-infected keratinocytes showed the formation of membrane protuberances, as well as actin focal points, starting 2 h post-infection, suggesting a reorganization of the cytoskeleton. These alterations were observed until 10 h post-infection.

Closer observation of infected cells at first showed that abundant actin focal points and membrane projections had formed at post-infection observation times. In addition, yeast-like structures were observed at the ends of the membrane projections (Figure 3). In contrast, no changes in the actin cytoskeleton were observed in the control group, in which stress fibers were observed.

### 3.3. Overexpression of TLR2, TLR6, MR and CR3 Receptors by Keratinocytes Infected with Conidia and Yeast Cells of Sporothrix schenckii

The keratinocytes express different PRRs on their cell surface [15,16,17,18,19,20], so we decided to analyze the expression of various cell receptors during infection with conidia and yeast cells of *S. schenckii*.

Keratinocytes infected with *S. schenckii* conidia (Figure 4A) showed an overexpression of the MR, TLR2, CR3, and TLR6 receptors starting 2 h post-infection for a maximum of 10 h, compared to the control group. At the same time, there was a discrete overexpression of TLR4. The results were confirmed by the mean fluorescence intensity analysis (Figure 4B), and the MFI for TLR2 had a significant value at 2, 6 and 10 h post-infection. In the case of TLR4, the MFI did not show a significant difference compared to the control group of uninfected cells, while in contrast, the cells treated with PMA showed a significant expression of TLR4. TLR6 expression reached significant MFI values at all times evaluated, with a maximum of 32 at 10 h. The MR and CR3 receptors showed significant differences at each of the times with a maximum MFI of 29 and 21, respectively, at 10 h post-infection. After a 24-h stimulation with PMA, the cells showed overexpression of TLR2, TLR4, TLR6, MR, and CR3.

*S. schenckii* yeast infection kinetics (Figure 5A) found an increase in the production of TLR6, MR, CR3, and TLR2 receptors starting 2 h post-infection, with a maximum expression at 10 h, compared to the control group of uninfected cells. The TLR4 expression was very discrete at the same post-infection times. The mean fluorescence intensity analysis (Figure 5B) confirmed the observations, finding a maximum overexpression of TLR2 at 10 h post-infection with an MFI of 15; while the maximum MFI for TLR6 was 37, the MR had a maximum value of 52, and the CR3 had a maximum value of 31. In the case of TLR4, no significant elevation of MFI was found with respect to the control group, except for the MFI of cells treated with PMA.

### 3.4. Infection of Keratinocytes with Sporothrix schenckii Induces the Production of Proinflammatory Cytokines, Chemokines and Growth Factors

To determine whether infection by *S. schenckii* induces cytokine production, infection kinetics were performed with *Sporothrix schenckii* conidia as described above. A total of thirty elements were determined, including cytokines, chemokines, and growth factors to be evaluated with the LUMINEX system, in the keratinocyte culture supernatants at each post-infection time. All the elements analyzed were described in the methodology section, and only the molecules whose production increased during the infection are presented.

The chemokines evaluated were RANTES, MCP-1, IL-8, and IP10. The keratinocytes infected with *Sporothrix schenckii* conidia showed a significant increase in the production of MCP-1 and IL-8. Production of MCP-1 and IL-8 started at 8 h, and reached their maximum concentration at 12 h, attaining levels of 390 pg/mL and 270 pg/mL, respectively. The keratinocytes produced RANTES significantly at 12 h post-infection (10 pg/mL). The production of the IP-10 chemokine was discrete and late at 12 h, reaching a maximum level of 1.5 pg/mL that was not statistically significant (Figure 6A).

On the other hand, during infection with conidia, production of two growth factors, granulocyte colony-stimulating factor (G-CSF) and granulocyte-macrophage colony-stimulating factor (GM-CSF), was observed. Maximum levels of G-CSF and GM-CSF were 61 pg/mL and 12 pg/mL, respectively (Figure 6C). The proinflammatory cytokines produced by the infected keratinocytes were IL-6 with 5.0 pg/mL, TNF-α with 2.5 pg/mL, IFN-γ with 6.4 pg/mL, and IFN-α with 7.0 pg/mL. These cytokines reached their maximum levels at 12 h post-infection (Figure 6B). Interestingly, conidial infection also stimulated the production of the anti-inflammatory cytokine IL-10 with a significant and elevated level of 27 pg/mL at 12 h (Figure 6B). In summary, the highest levels of soluble mediators produced by keratinocytes infected with *S. schenckii* conidia were the chemokines MCP-1 and IL-8, followed by G-CSF, IL-10, GM-CSF, IFN-α, IFN-γ, IL-6, and TNF-α, in decreasing order (Figure 6).

With regard to the production of mediators produced by keratinocytes infected with *Sporothrix schenckii* yeast cells, the chemokines analyzed were RANTES, MCP-1, IL-8, and IP10. The MCP-1 and IL-8 chemokines were significantly elevated, which were produced in high amounts in late observation times, reaching a maximum concentration of 480 pg/mL for MCP-1, and 407 pg/mL for IL-8, at a post-infection time of 12 h. However, no important or significant increase was observed for RANTES, which reached a maximum concentration of 10 pg/mL towards the end of the infection kinetics. The IP-10 chemokine, on the other hand, was produced late, reaching very low levels; i.e., a concentration of just 2 pg/mL at 12 h post-infection (Figure 7A).

During infection with yeasts, late production of G-CSF and GM-CSF was observed with a concentration of 62 pg/mL and 11 pg/mL, respectively. Likewise, the keratinocytes infected with yeasts produced proinflammatory cytokines. Those that significantly increased their levels included IL-6 with 7.5 pg/mL, IFN-α with 7 pg/mL, and IFN-γ with 6 pg/mL. All of these cytokines reached their maximum level at 12 h post-infection. On the other hand, TNF-α had a concentration of 1.9 pg/mL at 12 h post-infection, without there being a significant increase. Furthermore, infected keratinocytes also produced elevated levels of the anti-inflammatory cytokine IL-10, reaching a concentration of 28.8 pg/mL at 12 h.

As mentioned above, thirty molecules produced by keratinocytes infected with both phases of the *S. schenckii* fungus were evaluated, including chemokines, cytokines, and growth factors. Figure 8 is a heat map showing the production of these molecules at different times, following infection with conidia and yeasts of the fungus. It was observed that when infected with both phases of the fungus, keratinocytes significantly produced the chemokines IL-8 and MCP-1. G-CSF, IL-10, and GM-CSF were also produced to a lesser degree, and the cytokines IL-6, IFN-γ, IFN-α at a much lower concentration. In contrast, TNF-α production was significant only with conidial infection.

## 4. Discussion

Sporotrichosis is a mycosis that affects epidermis, dermis, and subcutaneous tissue, and it is caused by a thermally dimorphic fungal species of the complex *Sporothrix* spp. One of its etiological agents is the fungus *Sporothrix schenckii* [1]. A transition from mycelium to yeast is required for infection to settle in tissue, which successfully occurs among the species of the clinical clade [22]. Histologically, the lesions are pyogenic and granulomatous, showing infiltration of neutrophils, scarce eosinophils, mononuclear phagocytes, lymphocytes, and plasma cells. In addition, asteroid bodies and yeast cells are observed [23].

As the largest organ of the human body, the skin not only functions as a physical barrier, but also provides defense against these microorganisms [24]. Keratinocytes are among the epidermal cell lines that maintain the integrity of this barrier and tissue homeostasis [25]. In addition, when coming into contact with a microorganism, they participate by mediating antimicrobial responses, promoting a pro-inflammatory environment, and producing antimicrobial peptides in infections with actinomycetes, viruses and fungi [26,27,28]. However, information is limited on the involvement of keratinocytes in the pathogenesis of fungi such as *S. schenckii*.

In this study, we evaluated the keratinocyte response during infection with conidia (infective phase) and yeast cells (parasitic phase) of *S. schenckii*. Previous studies have shown differences in the virulence levels of the *Sporothrix* spp. complex, attributed to the thermotolerance and production of melanin that confers resistance to antifungals such as amphotericin B and terbinafine [29,30].

Our results showed that keratinocytes are susceptible to infection with conidia and yeast cells of *S. schenckii*, causing a certain degree of cell death (Figure 1). To date, there are no in vitro cytotoxicity studies of *Sporothrix* spp.

The most frequently reported virulence studies are in murine models, where it has been observed that *Sporothrix schenckii* shows different levels of virulence, and that this depends on the amount of inoculum administered [31,32,33]. Our study was conducted with a multiplicity of infection (MOI) of 1:1, and despite the low microbial load, the percentage of viable cells was affected, albeit at a low proportion. Another factor to consider is the origin of the clinical isolate. It has been reported that in the murine model, isolates from patients with disseminated sporotrichosis lead to a more severe disease compared to isolates from patients with lymphocutaneous sporotrichosis [34]. The clinical isolate we used in this study was from a patient with disseminated sporotrichosis; thus, the cytotoxic effect observed could be attributed to its clinical origin.

On the other hand, the infection of the host cell by the pathogenic fungus begins with its adhesion to the cell surface, and this interaction is fundamental for the pathogenesis of mycoses [35]. Previous studies have shown that opportunistic fungi such as *C. glabrata* adhere to the cell surface of human osteoblasts and induce the polymerization of actin filaments, which cause the formation of membrane projections that trap yeasts that get internalized by cells [36].

It has also been described that *S. schenckii* interacts with epithelial cells, leading to their morphological alteration and the loss of rearrangement of the microtubular network [37]. However, the role of the actin cytoskeleton in the internalization of *S. schenckii* in keratinocytes remains unknown. We found that infection with conidia and yeast cells of *S. schenckii* induced changes in the cell morphology of keratinocytes (Figure 2). These changes consisted of the reorganization in the polymerization of actin filaments, the formation of cellular projections, and the loss of stress fibers (Figure 3). These results are consistent with those produced by other pathogenic fungi such as *Malassezia pachydermatis*, *Aspergillus fumigatus*, *Paracoccidioides brasiliensis*, and *Cryptococcus neoformans* that induce rearrangement of the actin cytoskeleton in non-phagocytic cells [38,39,40,41]. The rearrangement of the actin filaments in keratinocytes and the formation of membrane protrusions could suggest the internalization of conidia and yeast cells of *S. schenckii* as a possible mechanism to infect the host cell. The changes observed are similar to those that *M. tuberculosis* induces upon being internalized in lung epithelial cells, through a mechanism of macropinocytosis [42]. The possibility that *S. schenckii* triggers the macropinocytosis mechanism to be internalized by keratinocytes is a fact that must be explored.

Keratinocytes contribute to the inflammatory process by producing mediators of the innate immune system during infectious processes [21]. The expression of these effector molecules begins by recognizing pathogen-associated molecular patterns (PAMPs) through the different pattern-recognition receptors (PRRs) [21]. Some components of the fungal cell wall, such as chitin, mannans and β-glucans are recognized by the PRRs [43,44]. The cell wall of the *S. schenckii* conidia is composed of rhamnose and mannose, whereas the yeasts have rhamnomanian peptides. The polysaccharide moiety of these rhamnomanian peptides is composed of D-mannose, L-rhamnose, and galactose polysaccharides [45,46], and it is unknown whether the PRRs can recognize them in keratinocytes.

The expression of PRRs by keratinocytes during *S. schenckii* infection is also not known in detail. Our results showed that keratinocytes infected with conidia overexpress MR, TLR6, CR3 and TLR2 (Figure 4). Similarly, in infection by yeast cells, MR, TLR6, CR3 and TLR2 receptors are overexpressed (Figure 5). High MR expression and low TLR4 expression were observed in both infections. Mannose receptor overexpression plays an important role in antifungal response [47]. This receptor recognizes mannosyl-fucosyl ligands and glycoconjugates present in fungi and is part of the C-type lectin receptors (CLRs) [47,48]. It is expressed in dendritic cells, macrophages, and also in human keratinocytes [48,49]. Therefore, overexpression of this receptor in keratinocytes during *S. schenckii* conidia and yeast infection could indicate that it actively participates in pathogen recognition and infection control, as has been described in *Candida albicans* infection [49].

On the other hand, CR3 is known to be involved in antifungal response, and is essential for the phagocytosis of particles opsonized by the iC3b complement fragment [50]. Studies have shown that *Histoplasma capsulatum* enters macrophages through this receptor [51]. Furthermore, previous research on THP-1 macrophages showed that they are capable of phagocytizing opsonized yeasts through CR3, and opsonized and non-opsonized conidia of *S. schenckii* through the MR [11]. Our results showed a high expression of CR3 during infection with both phases of the fungus. Most likely, this receptor is involved in the internalization of opsonized conidia and yeasts, although its role in the internalization of these pathogens has been described in phagocytic cells [11]. Similarly, the signaling pathways that the MR and CR3 could trigger in keratinocytes that lead to cell activation resulting in an antifungal state by the keratinocyte and/or the production of cytokines and chemokines are unknown. Research has shown that *Pneumocystis* stimulates nuclear NF-kB translocation through MR activation in alveolar macrophages, and the activation of this pathway is dependent on the multiplicity of infection [52]. In contrast, blocking the MR suppressed NF-kB expression in mast cells infected with *Bordetella pertussis* [53].

Similarly, activation of the NF-kB signaling pathway is also mediated by TLRs. The involvement of TLR2 and TLR4 in the interaction with conidia and yeast cells of *Sporothrix schenckii* in keratinocytes and their consequent activation of NF-kB has been demonstrated [54]. This activation triggers an inflammatory response mediated by IL-6 and IL-8 [54]. In contrast, our results showed a non-significant expression of TLR4, and an overexpression of TLR2 and TLR6. Both receptors can form heterodimers that recognize zymosan, i.e., mannan particles containing β (1,3)-glucan found in fungal cell walls [55]. The TLR2 and TLR6 heterodimer could recognize zymosan in the cell wall of *S. schenckii*.

In our study, the recognition of *S. shenckii* conidia and yeasts by keratinocytes activated signaling pathways that resulted in the production of the proinflammatory cytokines IL-6, TNF-α and IFN-α, IFN-γ; anti-inflammatory cytokine IL-10; chemokines IL-8, MCP-1 and growth factors GM-CSF, G-CSF (Figure 6, Figure 7 and Figure 8). TLR2 expression is involved in the production of IL-6 and IL-8 that promote the generation of an inflammatory state [54]. Additionally, during infection with both phases of the fungus, we observed not only elevated levels of IL-8, but also high levels of the MCP-1 chemokine. The production of MCP-1 can correlate with the cellular infiltrate of macrophages, neutrophils and eosinophils, observed in *S. schenckii* lesions, due to its chemoattractant function [56]. The results also showed low TNF-α production during infection with *S. schenckii* conidia, and no production during infection with yeasts. Synthesis of this cytokine has been shown to be through the MR via the P38 MAPK signaling pathway in human epithelial cells [57]. In addition, MR and TLR4 have been reported to be involved in the production of TNF-α and IL-6 [12]. Thus, low TLR4 expression could influence low TNF-α production during yeast infection.

The production of IFN-γ and IFN-α was significant during infection with both phases of the fungus. IFN-γ showed the most sustained production, and the synthesis of both types of interferons has been reported in inflammatory processes in keratinocytes [58,59]. Although the production of IFN-γ has been described to be mainly limited to immune response cells such as T cells, macrophages or NK cells [60], it is interesting that in the model of infection by both phases of the fungus, keratinocytes preferentially produce it over IFN-α. As such, the implications of this situation should be studied in greater detail. IFN-γ has an immunomodulatory function and is synthesized through the interferon regulatory factor (IRF) [60]. It is not known whether IRF is involved in the production of this cytokine by keratinocytes.

A relevant data in this model of infection is the production of IL-10 by keratinocytes in late post-infection times with yeasts or conidia. IL-10 is a primary cytokine in the modulation of inflammatory responses, in addition to regulating the growth of various cell lines including keratinocytes [61]. In human peripheral blood mononuclear cells, the production of IL-10 has been evaluated during infection with conidia and yeast cells of *S. schenckii* [12]. Production of this cytokine has been reported through the activation of the Dectin-1, TLR2 and MR receptors [12]. The presence of IL-10 from infected keratinocytes may contribute to an environment that favors infection by interfering with proper cellular immune response, although it can also contribute to containing the damage produced by hyperinflammation [62]. Further in vivo studies are necessary to establish the role of IL-10 in sporotrichosis.

Like the other mediators analyzed, a significant and late production of two growth factors was observed: GM-CSF and G-CSF. GM-CSF stimulates the differentiation and proliferation of macrophages, eosinophils, and granulocytes. It also induces the migration and proliferation of keratinocytes by stimulating the healing process [63]. G-CSF is an important factor in the proliferation and differentiation of neutrophils [64]. To date, there is no report on the production of these growth factors in the infection of keratinocytes by *Sporothix* spp. However, the production of GM-CSF and G-CSF has been reported in keratinocytes treated with dinitrochlorobenzene, and they have been attributed a pro-inflammatory effect [65], which could be contributing to the pathogenesis of Sporotrichosis.

## 5. Conclusions

Overall, this study shows the responsiveness of keratinocytes during early infection with conidia and yeast cells of *S. schenckii*. It begins with recognition by keratinocyte receptors including MR, CR3, TLR 6, and TLR2 for both phases of the fungus. After recognition, the keratinocyte undergoes changes in its cytoskeleton that induces the formation of membrane protrusions that can facilitate the internalization of conidia or yeasts. The infection promotes the production of cytokines, creating a pro-inflammatory, and, above all, chemotactic environment with very high production of MCP-1 and IL-8. This chemotactic environment will be responsible for the recruitment of other cell lines at the infection site. Whereas in late post-infection times keratinocytes produce IL-10, which could mediate an anti-inflammatory response and therefore aid in the survival of the pathogen, it could also eventually contribute to a protective effect for the host by decreasing hyperinflammation. At the same time, keratinocytes produce growth factors that could help repair damaged tissue during infection, or in the same way, contribute to the pro-inflammatory environment characteristic of the disease. Figure 9 depicts the hypothetical model of the findings of this study.

## Figures and Tables

**Figure 1 jof-08-00437-f001:**
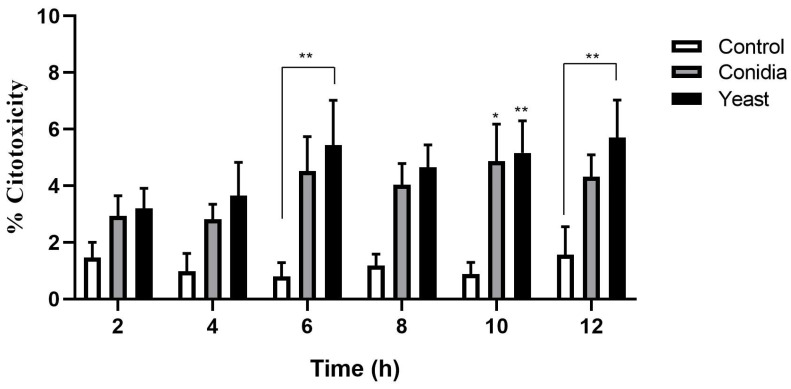
Cytotoxicity of keratinocytes during infection by conidia and yeast cells of *S. schenckii*. Cell infection kinetics were performed with conidia and yeast cells of *S. schenckii* in keratinocytes with a 1:1 MOI, for 12 h. At each post-infection time, the cell culture supernatants were recovered. The assessment was performed with the CytoTox 96^®^ Assay commercial kit using lactate dehydrogenase (LDH) detection. The data are presented as the mean ± standard deviation (SD) of three independent experiments. * *p* < 0.05, ** *p* < 0.005.

**Figure 2 jof-08-00437-f002:**
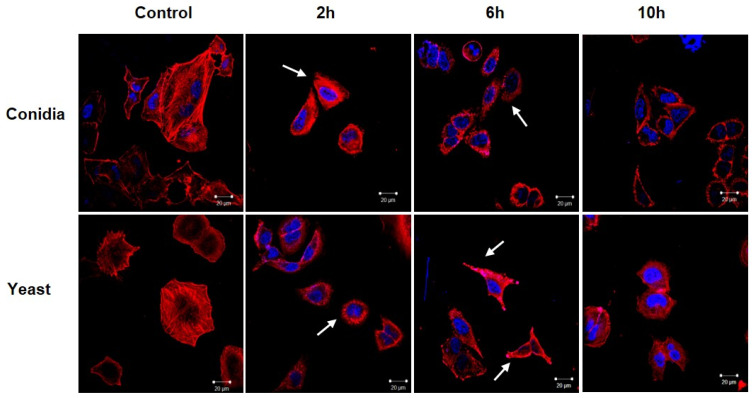
Changes in the cytoskeleton of keratinocytes during infection with conidia and yeast cells of *Sporothrix schenckii*. Cell infection kinetics were performed with conidia and yeast cells of *S. schenckii* in keratinocytes at a 1:1 MOI for 10 h. Actin filaments were stained with rhodamine phalloidin (red), and the cell nuclei with DAPI (blue). White arrows indicate the formation of membrane protuberances. Images in 60x.

**Figure 3 jof-08-00437-f003:**
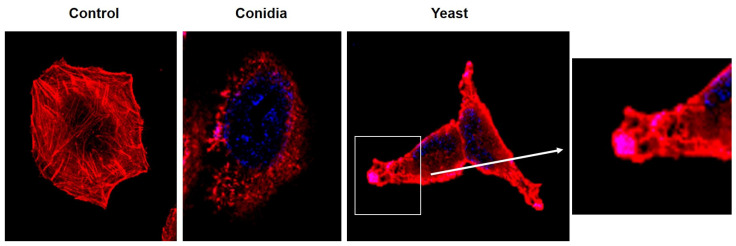
Formation of membrane protuberances and loss of stress fibers in keratinocytes infected with conidia and yeast cells of *Sporothrix schenckii*. Infection with conidia and yeast cells of *Sporothrix schenckii* at 6 h post-infection at a 1:1 MOI. The actin filaments were stained with rhodamine phalloidin (red), and the cell nuclei with DAPI (blue). Digitally amplified images.

**Figure 4 jof-08-00437-f004:**
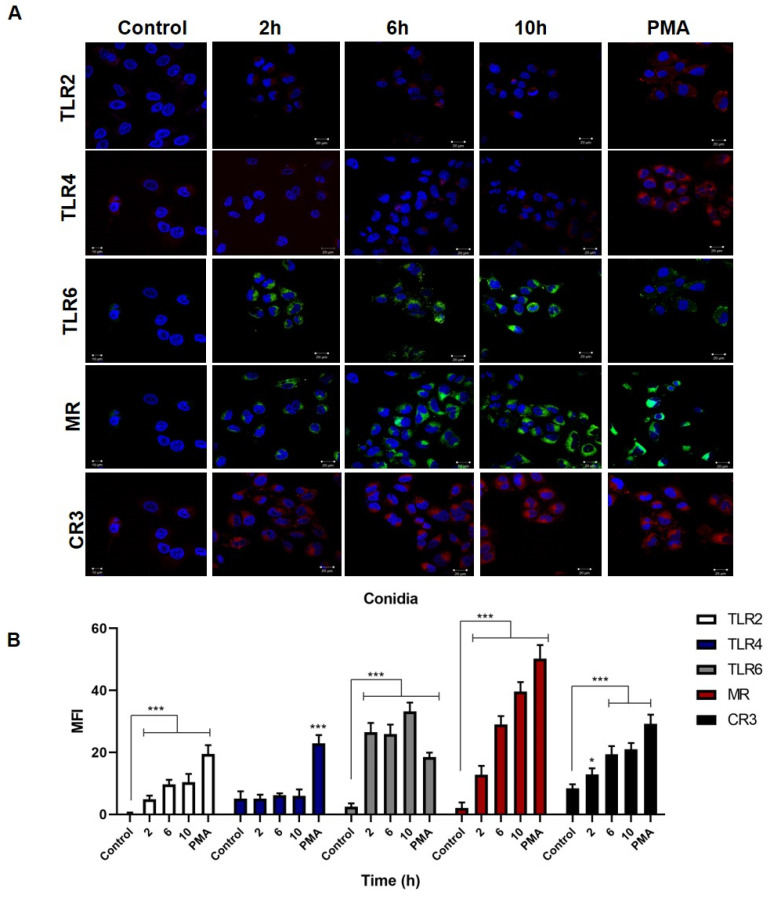
Expression of TLR2, TLR4, TLR6 receptors, the mannose receptor (MR) and the complement receptor (CR3) in keratinocytes infected with *S. schenckii* conidia. Cell infection kinetics were monitored for 10 h, and receptor expression was assessed by confocal microscopy. (**A**) Anti-TLR2, -TLR4, -TLR6, -MRC1, and -CR3 antibodies were used, incubated with mouse anti-IgG-FITC, mouse anti-IgG-TRITC, and cell nuclei with DAPI. (**B**) The mean fluorescence intensity (MFI) per cell was determined using the LSM5 application. Data are presented as the mean ± standard deviation (SD) of three experiments. * *p* < 0.01 and *** *p* < 0.0001.

**Figure 5 jof-08-00437-f005:**
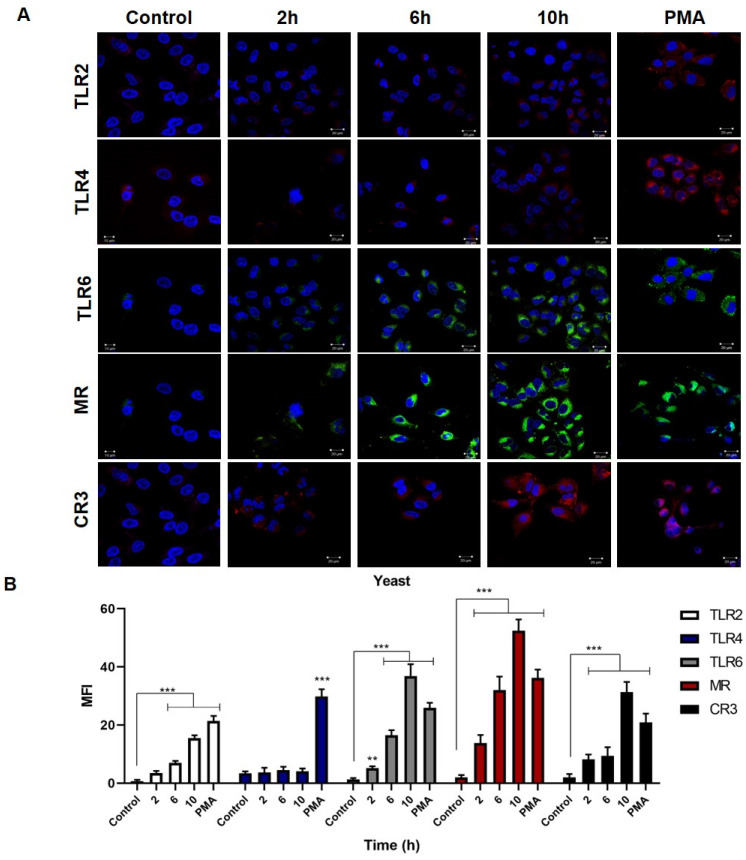
Expression of TLR2, TLR4, TLR6 receptors, the mannose receptor (MR) and the complement receptor (CR3) in keratinocytes infected with *S. schenckii* yeast cells. Cell infection kinetics were monitored for 10 h, and receptor expression was assessed by confocal microscopy. (**A**) Anti-TLR2, -TLR4, -TLR6, -MRC1, and -CR3 antibodies were used, incubated with mouse anti-IgG-FITC, mouse anti-IgG-TRITC, and cell nuclei with DAPI. (**B**) The mean fluorescence intensity (MFI) per cell was determined using the LSM5 application. Data are presented as the mean ± standard deviation (SD) of three experiments. ** *p* < 0.001 and *** *p* < 0.0001.

**Figure 6 jof-08-00437-f006:**
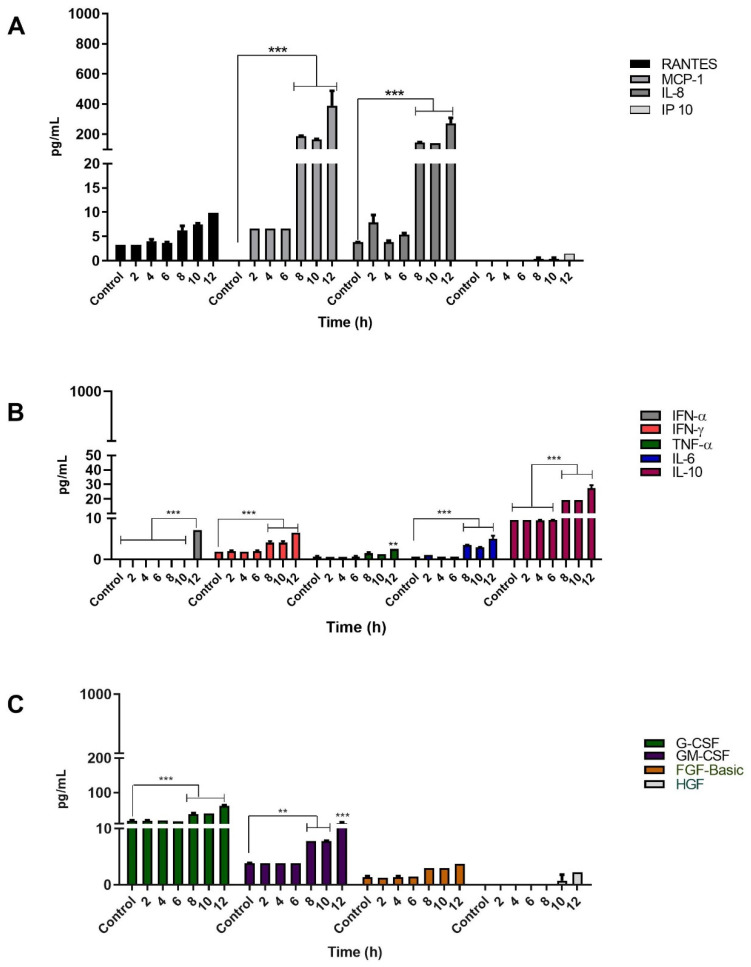
Cytokine production by keratinocytes infected with *S. schenckii* conidia. Infection Kinetics at 12 h, 1:1 MOI. Quantification of (**A**) chemokines, (**B**) cytokines and (**C**) growth factors was performed using LUMINEX technology on Magpix^®^ equipment following the instructions of the Cytokine Human Magnetic 30-Plex Panel for LuminexTM Platform (Thermo Fisher Scientific) kit. Data are presented as the mean ± standard deviation (SD) of two independent experiments. ** *p* < 0.001 and *** *p* < 0.0001.

**Figure 7 jof-08-00437-f007:**
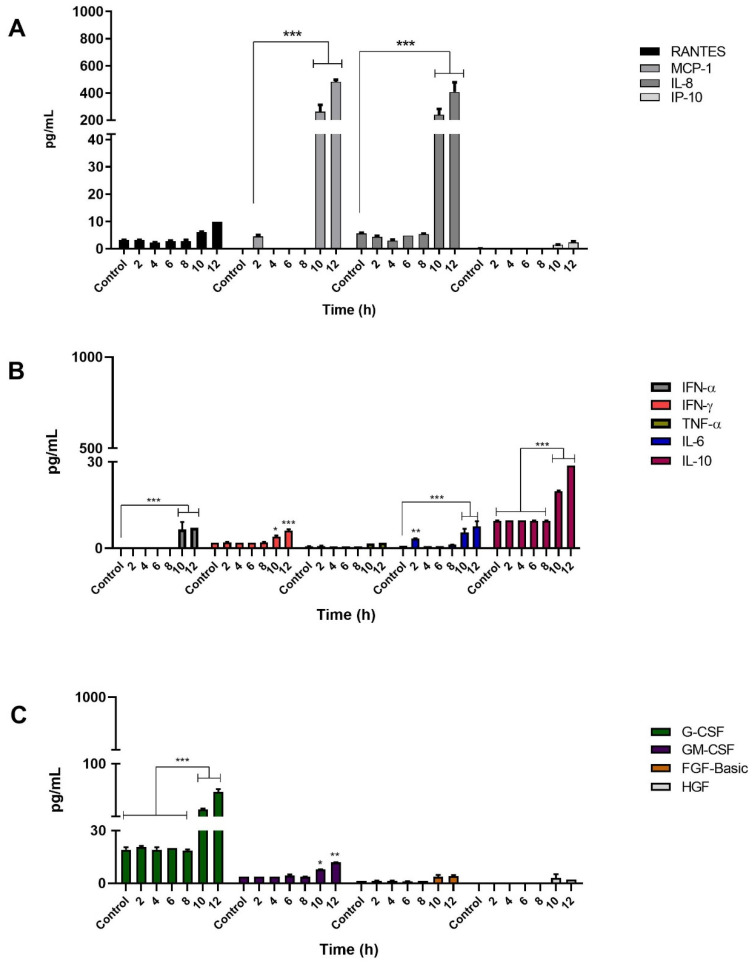
Cytokine production by keratinocytes (HaCaT) infected with *S. schenckii* yeast cells. Infection kinetics at 12 h, 1:1 MOI. Quantification of (**A**) chemokines, (**B**) cytokines and (**C**) growth factors was performed using LUMINEX technology on Magpix^®^ equipment following the instructions of the Cytokine Human Magnetic 30-Plex Panel for LuminexTM Platform (Thermo Fisher Scientific) kit. Data are presented as the mean ± standard deviation (SD) of two independent experiments. * *p* < 0.01, ** *p* < 0.001 and *** *p* < 0.0001.

**Figure 8 jof-08-00437-f008:**
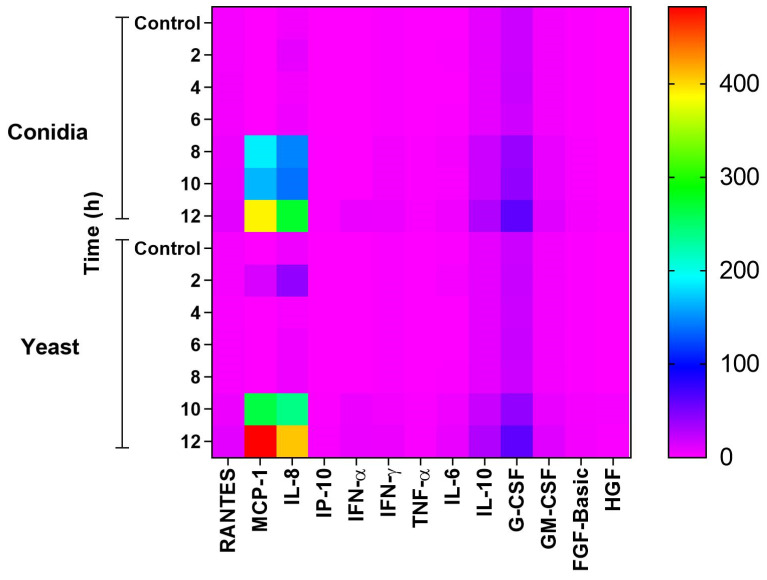
Heat map of cytokine, chemokine and growth factor production during infection of keratinocytes by conidia and yeast cells of *S. schenckii*. Production of different mediators was analyzed using the Luminex Technology. The supernatants of the keratinocytes infected with conidia and yeasts of the fungus *S. schenckii* were used at different post-infection times. The results are presented as the mean of two independent experiments of the concentration values (pg/mL) of each mediator analyzed.

**Figure 9 jof-08-00437-f009:**
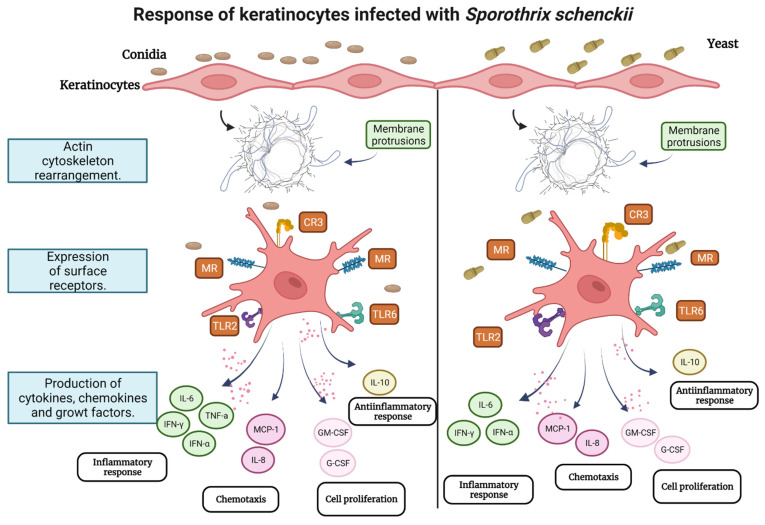
Hypothetical model of keratinocyte response during infection with conidia and yeast cells of *S. schenckii*.

## Data Availability

The data presented in this study are available on request.

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
