# Peer review of "Keratinocyte Response to Infection with Sporothrix schenckii"

_jof, 2022, doi:10.3390/jof8050437_

Round 1
Reviewer 1 Report
This is a well done study.
Only some minor suggestions:
- Introduction: sporotrichosis involves epidermis, dermis and subcutaneous tissue.
- Images of figure 4 should be improved.
- Figure 8 should be improved.
- Figure 9: please cite the source (if the image is not provided by the authors).
- Discussion is pointlessly long.
Author Response
Q1-Introduction: sporotrichosis involves epidermis, dermis and subcutaneous tissue
R1-We appreciate your observation, we modified the manuscript accordingly. We modified lines 41 and 406.
Q2-Images of figure 4 should be improved.
R2- We improved figure 4 changing it from the TIF format to JPG format, this modification improves the resolution.
Q3-Figure 8 should be improved.
R3-We improved figure 8 changing it from the TIF format to JPG format, this modification improves the resolution.
Q4-Figure 9: please cite the source (if the image is not provided by the authors).
R4- Figure 9 was created by us based on the results gotten in this study. The figure was elaborated with the BioRender.com tool. We clarified in the acknowledgment section this issue.
Q5-Discussion is pointlessly long.
R5- It might seem that the discussion is very long, however, since the study covered various aspects of the keratinocyte response, these points were addressed in the discussion and we believe that omitting them would leave it incomplete.

Reviewer 2 Report
Dear authors,
I read your manuscript concerning the epithelial cells as immune cell system, just know from last century. The study of immunological role of skin cells has been largely studied especially in psoriasis and LL-37. The paper is well structured and simple but I've some concerns:
- you used a clinical isolate of Sporothrix schenckii. Why didn't you perform a control ATCC Sporothrix schenckii? clinical isolates present peculiaritis and are extremly usefull in clinical pratic but it's always important to compare it with a standard.
-
as mentioned above, the immune role of skin epithelial cells has been known for several years, allowing us to understand that among the most important molecules governing epithelial-mediated immunity there are retinoids, derivatives of vitamin A. In fact, they not only act on innate and adaptive immunity, but are essential for correct epidermal differentiation, being effective both directly and indirectly in fungal infections. Read and cite:- Cosio T, Gaziano R, Zuccari G, Costanza G, Grelli S, Di Francesco P, Bianchi L, Campione E. Retinoids in Fungal Infections: From Bench to Bedside. Pharmaceuticals (Basel). 2021 Sep 24;14(10):962. doi: 10.3390/ph14100962. PMID: 34681186; PMCID: PMC8539705.- Campione E, Cosio T, Lanna C, Mazzilli S, Ventura A, Dika E, Gaziano R, Dattola A, Candi E, Bianchi L. Predictive role of vitamin A serum concentration in psoriatic patients treated with IL-17 inhibitors to prevent skin and systemic fungal infections. J Pharmacol Sci. 2020 Sep;144(1):52-56. doi: 10.1016/j.jphs.2020.06.003. Epub 2020 Jun 11. PMID: 32565006.
Author Response
Q1- You used a clinical isolate of Sporothrix schenckii. Why didn't you perform a control ATCC Sporothrix schenckii? clinical isolates present peculiaritis and are extremly usefull in clinical pratic but it's always important to compare it with a standard.
R1- We decided to use a clinical isolate of Sporothrix schenckii recovered from a patient diagnosed with disseminated sporotrichosis identified by microbiological and molecular methods by one of the authors of this work (RAG), and we did not use the ATCC strains since, according to their respective technical sheets published by the ATCC itself, all those consulted were classified as clinical isolates and not reference strains.
Q2- As mentioned above, the immune role of skin epithelial cells has been known for several years, allowing us to understand that among the most important molecules governing epithelial-mediated immunity there are retinoids, derivatives of vitamin A. In fact, they not only act on innate and adaptive immunity, but are essential for correct epidermal differentiation, being effective both directly and indirectly in fungal infections. Read and cite:- Cosio T, Gaziano R, Zuccari G, Costanza G, Grelli S, Di Francesco P, Bianchi L, Campione E. Retinoids in Fungal Infections: From Bench to Bedside. Pharmaceuticals (Basel). 2021 Sep 24;14(10):962. doi: 10.3390/ph14100962. PMID: 34681186; PMCID: PMC8539705.- Campione E, Cosio T, Lanna C, Mazzilli S, Ventura A, Dika E, Gaziano R, Dattola A, Candi E, Bianchi L. Predictive role of vitamin A serum concentration in psoriatic patients treated with IL-17 inhibitors to prevent skin and systemic fungal infections. J Pharmacol Sci. 2020 Sep;144(1):52-56. doi: 10.1016/j.jphs.2020.06.003. Epub 2020 Jun 11. PMID: 32565006.
R2- We agree with the reviewer that skin cells have been studied for several years, however, the details of the response of keratinocytes as specialized skin cells to S. schenckii infection are not known. Most of the studies of the response of epithelial cells of the skin have been carried out in non-infectious diseases, some of them of an autoimmune type such as psoriasis (https://pubmed.ncbi.nlm.nih.gov/?term= keratinocyte+psoriasis+hacat&sort=pubdate).
Other studies of keratinocyte response have focused on infectious processes caused by bacteria such as S. aureus (https://pubmed.ncbi.nlm.nih.gov/?term=keratinocyte+s.+aureus+hacat&sort=pubdate ), viruses like Herpes (https://pubmed.ncbi.nlm.nih.gov/?term=keratinocyte+herpes+hacat&sort=pubdate), and even signs of damage like UV light (https://pubmed.ncbi.nlm .nih.gov/?term=keratinocyte+herpes+hacat&sort=pubdate).
However, the response of these cells to fungal infection, particularly S. schenckii, is scarce (https://pubmed.ncbi.nlm. nih.gov/?term=keratinocyte+herpes+hacat&sort=pubdate), which is why our work is an original contribution and one of the first reports to address this issue.
Regarding the cites referred by the reviewer, we analyzed them in depth and although they are interesting, both works are not directly related to the central objective of our work, which is the description of the innate response of keratinocytes to infection by S. schenckii, it is initially of interest to us to describe the response mechanisms of keratinocytes to this infection, to later search for new treatment strategies such as those described in the articles suggested by the reviewer.
Reviewer 3 Report
no comments for the Authors
Author Response
Q1- No comments for the Authors
R1- We appreciate the review of our manuscript.
Round 2
Reviewer 2 Report
Dear Authors,
all concerns have been addresed.